# Arbuscular Mycorrhizal Symbiosis Primes Tolerance to Cucumber Mosaic Virus in Tomato

**DOI:** 10.3390/v12060675

**Published:** 2020-06-22

**Authors:** Laura Miozzi, Anna Maria Vaira, Federico Brilli, Valerio Casarin, Mara Berti, Alessandra Ferrandino, Luca Nerva, Gian Paolo Accotto, Luisa Lanfranco

**Affiliations:** 1Institute for Sustainable Plant Protection, National Research Council of Italy (IPSP-CNR), Torino, Strada delle Cacce 73, 10135 Torino, Italy; annamaria.vaira@ipsp.cnr.it (A.M.V.); valeriocasarin@yahoo.it (V.C.); bertimara@gmail.com (M.B.); luca.nerva@crea.gov.it (L.N.); Gianpaolo.Accotto@ipsp.cnr.it (G.P.A.); 2Institute for Sustainable Plant Protection, National Research Council of Italy (IPSP-CNR), Unit of Sesto Fiorentino (FI), Via Madonna del Piano 10, 50019 Sesto Fiorentino (FI), Italy; federico.brilli@ipsp.cnr.it; 3Department of Agricultural, Forestry and Food Sciences, University of Torino, Largo Paolo Braccini 2, 10095 Grugliasco (TO), Italy; alessandra.ferrandino@unito.it; 4Council for Agricultural Research and Economics—Research Centre for Viticulture and Enology CREA-VE, Via XXVIII Aprile 26, 31015 Conegliano (TV), Italy; 5Department of Life Sciences and Systems Biology, University of Torino, Viale Mattioli 25, 10125 Torino, Italy

**Keywords:** arbuscular mycorrhizal symbiosis, cucumber mosaic virus, *Funneliformis mosseae*, gene expression, priming tolerance, plant–virus interaction, RNA sequencing, *Solanum lycopersicum* L.

## Abstract

Tomato plants can establish symbiotic interactions with arbuscular mycorrhizal fungi (AMF) able to promote plant nutrition and prime systemic plant defenses against pathogens attack; the mechanism involved is known as mycorrhiza-induced resistance (MIR). However, studies on the effect of AMF on viral infection, still limited and not conclusive, indicate that AMF colonization may have a detrimental effect on plant defenses against viruses, so that the term “mycorrhiza-induced susceptibility” (MIS) has been proposed for these cases. To expand the case studies to a not yet tested viral family, that is, Bromoviridae, we investigated the effect of the colonization by the AMF *Funneliformis mosseae* on cucumber mosaic virus (CMV) infection in tomato by phenotypic, physiological, biochemical, and transcriptional analyses. Our results showed that the establishment of a functional AM symbiosis is able to limit symptoms development. Physiological and transcriptomic data highlighted that AMF mitigates the drastic downregulation of photosynthesis-related genes and the reduction of photosynthetic CO_2_ assimilation rate caused by CMV infection. In parallel, an increase of salicylic acid level and a modulation of reactive oxygen species (ROS)-related genes, toward a limitation of ROS accumulation, was specifically observed in CMV-infected mycorrhizal plants. Overall, our data indicate that the AM symbiosis influences the development of CMV infection in tomato plants and exerts a priming effect able to enhance tolerance to viral infection.

## 1. Introduction

Viruses are ubiquitous obligate intracellular parasites able to cause severe diseases worldwide [1,2,3,4]. Virus infections dramatically impair host plant performances by hampering photosynthetic carbon assimilation [5] and alter a number of processes within plants, such as cell cycle, transport, protein modifications, secondary metabolism, hormone regulation, and biosynthesis of volatile and nonvolatile isoprenoids, thus weakening plant defenses and promoting virus transmission and replication [6]. To date, genetic resistance and control of vectors (insect/fungi and others) are the most widespread strategies against viral infection. However, due to the viral high mutation and recombination rates [7,8,9], only a limited number of effective plant resistance genes are available and the acquisition of genetic resistance through crossing is not always obtainable [10]. Moreover, physical or chemical methods aimed to limit viral vector populations, such as physical barriers and chemical insecticides, are useful but not decisive in contrasting virus infection and may have negative consequences on nontarget organisms, public health, and environment [11,12,13].

Cucumber mosaic virus (CMV, type species of the genus *Cucumovirus*, family Bromoviridae) is an economically important RNA virus belonging to the top 10 list of plant viral pathogens [2,14] and has major agricultural impact worldwide. CMV is characterized by a wide host range including about 1200 species in more than 100 plant families; it is horizontally transmitted by many species of aphids in a nonpersistent manner and in some cases also by seed with different efficiencies. Symptoms vary with the host species and/or CMV strain and include chlorosis, necrosis, dwarfing, and leaf malformation. The viral genome consists of three genomic and two subgenomic RNAs. Genomic properties and virulence determinants of CMV have been reviewed in [15].

Arbuscular mycorrhizal fungi (AMF), able to colonize the roots of many plant species, including food crops [16], have recently gained increasing attention as a cost-effective and sustainable solution to improve the nutritional status of the plants and help them to cope with abiotic and biotic stresses in both marginal lands and traditional organic agricultural systems [17,18,19]. In addition to promoting plant mineral nutrition, the establishment of a functional AM symbiosis may elicit molecular and physiological mechanisms of plant signaling and prime systemic plant defenses that enhance tolerance to environmental stresses [20,21,22,23] and pathogens attack [24]. The boost of basal defenses in mycorrhizal plants is defined as priming [25], and the consequent mycorrhiza-induced resistance (MIR) has been defined as a cumulative effect of plant responses to mycorrhizal colonization, able to confer protection against a wide range of challengers, including biotrophic and necrotrophic pathogens, nematodes, and insects [26,27].

It has been shown that AMF colonization leads to a genome-wide transcriptional reprogramming of primary metabolism [28,29] that stimulates photosynthetic carbon assimilation and biomass production [30] and improves growth and productivity of plants under abiotic stress conditions [31]. The establishment of a functional AM symbiosis also triggers deep changes in plant secondary metabolism, including variations of phenolic compound content [29,30,31,32] and alterations in the accumulation of phytohormones such as jasmonic acid (JA), salicylic acid (SA), abscisic acid (ABA) [33,34].

Although AMF colonization has been already shown to decrease the severity of diseases caused by several plant pathogens [24], the effects of AMF on plant–virus interaction have received limited consideration and are not yet fully understood. The final outcome of a plant–AMF–virus tripartite interaction relies on several factors, including viral pathogen lifestyle, nutritional status of the plant, and timing of interaction [35]. The term “mycorrhiza-induced susceptibility” (MIS) has been proposed for the cases where AMF colonization increases the susceptibility of plants to viruses [35].

In the present work, we expand the case studies to a not yet considered family of plant viruses, the Bromoviridae, by evaluating the impact of colonization by *Funneliformis mosseae* (Syn. *Glomus mosseae*), an AMF widely distributed in agricultural and natural ecosystems, on infection by CMV in tomato (*Solanum lycopersicum* L.), one of the most important crops worldwide and a model system for plant–pathogen interactions. To this aim, we carried out analyses at physiological, biochemical, and molecular level.

## 2. Materials and Methods

### 2.1. Biological Materials, AMF Inoculation, and Virus Infection

Tomato seeds (cv. Moneymaker) were surface-sterilized by washing in 70% ethanol with a few drops of Tween 20 for 3 min, 5% sodium hypochlorite for 13 min, three times in distilled water for 10 min each time. The seeds were placed in Petri dishes containing water agar (0.6%), incubated for 1 week in the dark (25 °C), and then exposed to light for another week with a photoperiod of 14/10 h light/dark. Seedlings were then transferred to pots containing sterile quartz sand. Four groups of twelve plants each were established: control plants (C), CMV-infected plants (V), mycorrhizal plants (M), and CMV-infected mycorrhizal plants (MV). Plants were maintained in a growth chamber at 23 °C under a light intensity of 500 μE m^-2^ s^-1^ for 14 h followed by 10 h of dark, and watered twice a week: once with a modified Long Ashton nutrient solution [36] containing 320 μM phosphate and once with water. Phosphate content was optimized in the nutrient solution so that control plants did not suffer from phosphate limitations with respect to mycorrhizal ones [37]. 

Inoculation of *F. mosseae* Gerd. & Trappe isolate BEG12 (https://www.i-beg.eu/cultures/BEG12.htm) was performed by mixing an inoculum containing mycorrhizal root fragments of sorghum (MycAgro, France) with sterile quartz sand (30% *v/v*). Control plants were prepared by adding a blank inoculum composed by non-AMF sorghum roots fragments obtained in parallel and in the same conditions as those used to obtain the *F. mosseae* inoculum by MycAgro. To guarantee the establishment of arbuscular mycorrhiza prior to the challenge, CMV inoculation was performed 4 weeks following the AMF inoculation.

Plants were mechanically inoculated with CMV-FNY strain [38] on tomato leaves, previously dusted with silicon carbide, using a grinded sap extract from symptomatic leaves of CMV-infected *Nicotiana benthamiana* plants, diluted 1:10 (*w/v*) in the inoculation buffer (0.05 M K phosphate pH 7, plus 5 mM diethyldithiocarbamic acid (DIECA), 1 mM EDTA, and 5 mM Na-thioglycolate). Virus-free tomato plants underwent the same procedure, but mechanical inoculation was performed by using leaves extract from healthy *N. benthamiana* plants.

At 14 and 28 days post inoculation (dpi), the second and fifth leaves from the apex were collected, frozen in liquid nitrogen, and stored at –80 °C. The second leaves harvested at 14 and 28 dpi were used for viral genome quantification, while only the ones harvested at 28 dpi were used for RNA-Seq experiments. Apex, first, and second leaves harvested at 28 dpi were used for phytohormones quantification; the fifth leaf from the apex at 28 dpi, young but fully expanded, was used for physiological measurements. Plants were harvested at 28 dpi and weighed for biomass evaluation. One-way ANOVA with post hoc Tukey test was used to test statistical significance (*p* < 0.05).

### 2.2. Evaluation of Mycorrhizal Colonization, CMV Infection, and Viral Replication

For mycorrhizal evaluation, at 28 dpi plants were harvested and roots were stained overnight with 0.1% (*w/v*) blue cotton and washed with lactic acid. Randomly selected root segments were cut into 1 cm pieces and observed under a light microscope; mycorrhizal colonization was evaluated according to [39]. In particular, the following parameters were considered: M%, the intensity of the mycorrhizal colonization within each root fragment analyzed; F%, the number of colonized root fragments among those analyzed; a%, the arbuscule abundance in colonized fragments of the root; A%, the arbuscule abundance in the whole root system. The one-way ANOVA with post hoc Tukey test was used to evaluate statistical significance (*p* < 0.05).

At 14 and 28 dpi, CMV symptoms severity was estimated through visual inspection and, to evaluate CMV replication, quantitative RT-PCR (qRT-PCR) assays were performed on total RNA extracted from 100 mg of leaf tissue by Spectrum Plant Total RNA Kit (Sigma-Aldrich, MI, USA) according to the manufacturer’s instructions. RNA samples were treated with Turbo DNase free (Ambion, Foster City, CA, USA) according to the manufacturer’s instructions, and DNA absence was evaluated by PCR using 18S rRNA-specific primers. Quality and amount of RNAs were evaluated using NanoDrop spectrophotometer (Thermo Fisher Scientific, Waltham, MA, USA). Single-stranded cDNA was obtained from approximately 1500 ng of total RNA using the High-Capacity cDNA Reverse Transcription Kit (Thermo Fisher Scientific, Waltham, MA, USA), following manufacturer instructions. cDNA was amplified with primers CMV_CP(+)/(-) targeting the viral coat protein coding region, and UBC_418F/535R targeting the reference plant gene Solyc06g072570 encoding an ubiquitin conjugating enzyme (UBC) (Appendix A). The qRT-PCR assays were carried out using iTaq Universal SYBR Green Supermix (Bio-Rad, Hercules, CA, USA) in the CFX Connect™ Real-Time PCR Detection System (Bio-Rad, Hercules, CA, USA). Reactions were conducted in a total volume of 10 μL, containing 40 ng DNA, 5 μL SYBR Green Supermix (2×), and 0.3 μM of each primer. The PCR cycling program was as follows: 95 °C for 5 min, 40 cycles consisting of 95 °C for 10 sec each, 60 °C for 30 sec. A melting curve was recorded at the end of each run to assess amplification production specificity. For any reaction, three technical replicates were performed for each one of the three biological replicates. PCR efficiency was determined by standard curves constructed with serial dilutions of cDNA from CMV-infected tomato plants. The relative amount of CMV genome was calculated by using the Ct method according the following formula: 2-ΔΔCt, where ΔΔCt = ΔCtMV − ΔCtV [40].

### 2.3. Illumina Sequencing, Bioinformatic Analysis, and qRT-PCR Validation

The leaves harvested at 28 dpi were used for the RNA-Seq experiment. Three biological replicates were utilized for each condition. Each biological replicate was prepared by pooling RNA from leaves collected in three plants. For each biological replicate, 10 mg of total RNA was sent to Human Genetic Foundation sequencing service (http://www.hugef-torino.org/site/index.php) for library preparation and sequencing with Illumina TruSeq technology. A single-end sequencing with 75 bp of reads length was performed.

Raw data were checked for contaminants, and low-quality reads with fastQC software (http://www.bioinformatics.babraham.ac.uk/projects/fastqc). The spliced read mapper TopHat version 2.0.13 [41] was used to map reads on the tomato genome SL2.50 (ITAG2.4) obtained from the Solanaceae Genomic Network website (https://solgenomics.net). The Cufflinks suite was used for the transcriptome analysis [41]: Cufflinks assembler 2.2.1 was employed for assembling transcripts, Cuffcompare to check sensitivity and specificity of assemblies [42], the meta-assembler Cuffmerge 1.0.0 to merge transcripts and achieve a final transcriptome assembly. Differentially expressed genes (DEGs) were identified by using Cuffdiff 2.2.1 with a false discovery rate (FDR) threshold of 0.05. The R package CummeRbund 2.0.0 [41] was used to inspect, manage, and visualize the data. Details on parameters used for each software are specified in Appendix A.

Functional analysis of DEGs was performed using the MapMan tool (version 3.5.1R2) [43]. ITAG2.4 tomato transcriptome was used for annotation. Functional categories were tested for enrichment significance by using the Wilcoxon rank sum test included in MapMan software; BINS (MapMan functional categories) with a Benjamini Hochberg corrected *p*-value < 0.05 were considered statistically significant.

The qRT-PCRs were carried out to validate RNA-seq results, according to the protocol described before, using the primers reported in Appendix A.

### 2.4. Leaf Gas Exchange Measurements

Gas exchange measurements of CO_2_ and H_2_O were performed by using a portable gas exchange system (Li-COR6400, Li-COR Biosciences Inc., NE, USA) at 28 dpi. The terminal leaflet blade was clamped in the 6 cm^2^ LI-COR cuvette, and photosynthesis (A), stomatal conductance (gs), and substomatal CO_2_ concentration (Ci) were measured. During all the measurements, leaves were exposed to photosynthetic photon flux density (PPFD) of 500 μmol m^-2^ s^-1^, CO_2_ concentration of 400 ppm (achieved by fully scrubbing CO_2_ from ambient air with soda lime and replacing it with the LI-COR6400 CO_2_-injector system) under controlled temperature of 25 °C and relative humidity (RH %) ranging between 45% and 50%. Dark respiration was measured by switching off the light source within the leaf cuvette and waiting for 10 min of acclimation under the same conditions of temperature, RH %, and CO_2_ concentration. 

### 2.5. Hormone Measurements

Contents of ABA, indolacetic acid (IAA), and SA were quantified on three/four biological replicates as previously reported in [44]. Collected leaves at 28 dpi were lyophilized and then homogenized, samples were transferred in a 2 mL centrifuge tube (20–120 mg) and extracted with 1 mL of methanol:water (1:1 *v/v*) acidified with 0.1% of formic acid in an ultrasonic bath for 1 h. Samples were centrifuged at 15,000 rpm and 4 °C for 10 min, and the supernatant was analyzed by HPLC-DAD technique.

Original standard of ABA (purity ≥ 98.5 %), IAA (purity ≥ 99 %), and SA (purity ≥ 99 %), purchased from Sigma-Aldrich, were used for the identification of the studied metabolites by comparing retention times and UV spectra. The quantification was made by external calibration method. The HPLC apparatus was an Agilent 1220 Infinity LC system model G4290B (Agilent^®^, Waldbronn, Germany), equipped with gradient pump, autosampler, and column oven set at 30 °C. A 170 Diode Array Detector (Gilson, Middleton, The USA) set at 265 nm (for ABA and IAA) and 280 nm (for SA) was employed. A XTerra RP18 analytical column (150 × 4.6 mm i.d., 5 µm, waters) was used. The mobile phases consisted of water acidified with formic acid 0.1 % (A) and acetonitrile (B), at a flow rate of 0.500 mL min^-1^ in gradient mode, 0–20 min: from 10% to 35% of B, 20–25 min: from 35% to 100% B. For each sample, 20 μL were injected, and three biological replicates were run for each analysis.

## 3. Results

Our experimental design considered four different conditions: nonmycorrhizal and noninfected control plants, CMV-infected plants, mycorrhizal plants, and CMV-infected mycorrhizal plants. All the reported results refer to data obtained at 28 dpi, corresponding to 56 days post inoculation with mycorrhizal inoculum, except for viral symptomatology and viral titer that were evaluated at both 14 and 28 dpi.

### 3.1. Mycorrhizal Colonization and Viral Infection 

As a first step, we evaluated the AMF colonization of the roots, in order to be sure that all the plants were successfully colonized by the AMF and to evaluate a possible impact of viral infection on AMF intraradical growth (Appendix A). All M and MV plants, except one M plant that was discarded, showed a good mycorrhization level. At this time point, AMF colonization in MV plants, in terms of mycorrhization intensity and percentage of arbuscules within the entire root system, showed a slight increase when compared with the M plants. On the contrary, no statistically significant differences in mycorrhization frequency and number of arbuscules in the colonized segments of root were observed (Figure 1). Overall, these data indicate that the progression of viral infection has no drastic impact on *F. mosseae* intraradical growth.

The visual inspection and disease severity index (DSI)-based evaluation of viral symptomatology [45] highlighted that, at 14 dpi, viral symptoms in V plants were significantly more severe (leaf necrosis, severe leaf deformation, severe mosaic, and stunting, average DSI = 9.3) than in MV plants (mild mosaic, mild leaf deformation, average DSI = 3.0) (Figure 2A,B). Two weeks later, at 28 dpi, both V and MV plants showed similar mild mosaic and deformation on the new leaves, with an average DSI of 2.3. Interestingly, at both time points, the viral titer, as measured by qRT-PCR, was not statistically different between V and MV plants (Figure 2C), indicating that the mycorrhizal colonization influenced symptom development, but not virus accumulation. 

### 3.2. Biomass Production and Allocation and Leaf Gas Exchange

We further characterized our biological system by evaluating the hypogean and epigean biomass (Appendix A). In M plants, with respect to C plants, we observed a reduction of the hypogean biomass (Figure 3A); this may be linked to the fact that M plants rely for nutrient uptake on the efficiency of extraradical hyphae extending well beyond the root surface. A similar trend, even if not statistically significant, was observed in [39], where the same high phosphate concentration (320 µM) was used to grow the plants. In V plants, in comparison to C plants, we highlighted a slight tendency to the decrease of the epigean biomass and a drastic reduction of the hypogean biomass, possibly linked to the systemic spread of the virus in the roots system of the plant. In MV plants, biomass values were similar to those of V plants. (Figure 3A,B).

At the same time point, with respect to C plants, a decrease in the instantaneous photosynthetic CO_2_ assimilation rate was measured in leaves of V plants, while only a reduction trend was observed in the leaves of M and MV plants (Figure 4A upper part). Reduced photosynthetic CO_2_ assimilation rate correlated with the lower stomatal conductance observed in leaves of V, M, and MV compared with C plants (Figure 4B). Remarkably, in MV plants, *F. mosseae* colonization could alleviate the strong impact on stomatal conductance caused by the virus. The reduction of photosynthesis and stomatal conductance differently affected Ci (Figure 4C); in leaves of M and MV plants, both photosynthetic CO_2_ assimilation rate and stomata conductance decreased to the same extent, resulting in Ci values similar to those of C plants, while, in leaves of V plants, stomatal conductance dropped more than photosynthesis, thus reducing the Ci value. No differences were detected in CO_2_ respiration fluxes among the four groups of plants (Figure 4A, lower part). Row data are reported in Appendix A.

### 3.3. Hormones Measurement

Since SA is known to play a central role in plant response to pathogens [46], we measured its levels in our plants. The lowest levels of SA were measured in the leaves of C, M, and V plants, while a higher amount was detected in MV plants, suggesting that the concurrent presence of AMF and CMV may induce an increase in SA level (Figure 5A, Appendix A).

ABA has been recently involved in response to pathogens as well as in mycorrhiza and other compatible mutualistic interactions [47]. In respect to C plants, we observed a statistically significant increase of ABA in M plants, and only a trend to increase in V and MV plants (Figure 5B, Appendix A).

Beyond SA and ABA, plant defenses are also driven by JA and its balance with SA. Therefore, we measured the JA level in our plants; unfortunately, in our conditions it was under the detection limit in all four theses.

Beyond regulating plant development, IAA has been recently related to stress and defense responses through a crosstalk with other hormones [46]. With respect to C plants, our results highlighted a trend to the increase of IAA content in M plants and a tendency towards a reduction in V and MV plants (Figure 5C, Appendix A)

### 3.4. RNA-seq Data and Functional Characterization of DEGs

After quality and contaminant-free filtering, reads varied between 10 and 20 million per sample; for detailed statistics see Appendix A.

DEGs identified in respect to C plants were as follows: 4507 DEGs (2401 upregulated and 2106 downregulated) in V plants, 991 DEGs in M plants (645 upregulated and 346 downregulated), and 2385 in MV plants (1468 upregulated and 917 downregulated) (Figure 6A). The complete list of DEGs is reported in Appendix A. Overall, as showed by principal component analysis (PCA), CMV infection had the main impact on the transcriptome of V plants, while in M plants, AMF colonization caused a limited number of changes in gene expression (Figure 6B). An intermediate situation was observed in MV plants, owing to the contemporary presence of virus and AMF, with 42% of DEGs specific of this condition. These RNA-seq gene expression results were successfully validated by qRT-PCR analysis (Appendix A and Appendix A). 

MapMan analysis lead to the identification of 72 functional categories and subcategories significantly overrepresented in the case of viral infection (V plants), 32 in the case of contemporary presence of both the AMF and virus (MV plants), and only one category in the case of mycorrhization (M plants) (Appendix A). An overview of these functional categories is provided in Figure 6C.

*Primary metabolism.* Regulation of genes involved in the main processes of leaf primary carbon and energy metabolism highlighted that CMV infection had a strong impact on photosynthesis. In particular, genes related to the photochemical light-dependent reactions, as well as genes linked to mitochondrial electron transport, were strongly downregulated by CMV infection in V plants, compared with C plants, but not differentially regulated in M and MV plants (Appendix A), thus highlighting the buffering effect induced by the mycorrhization. In agreement with what was observed through physiological analyses, in MV plants, the AMF colonization shows a tendency to attenuate the reduction of the photosynthetic performance caused by CMV (Figure 4A, upper part and Appendix A). DEGs related to the major carbohydrate metabolism, and particularly to starch degradation, were overrepresented and mainly induced by the simultaneous presence of AMF and CMV (in MV plants), while no overrepresentation of these categories was observed in the other groups of plants (Appendix A and Appendix A).

*Phytohormones*. Considering hormone metabolism, an enrichment of DEGs related to ethylene biosynthesis was observed in V plants only. In particular, different isoforms coding for 1-aminocyclopropane-1-carboxylate synthase (ACCS) and 1-aminocyclopropane-1-carboxylate-oxidase (ACCO), both enzymes belonging to the ethylene biosynthetic pathway, were induced by CMV infection, while a general downregulation of these genes was measured in M plants. Finally, one ACCS and four ACCO coding genes were induced by the simultaneous presence of CMV and AMF (Appendix A and Appendix A). In V plants only, we found an overrepresentation of DEGs related to IAA and gibberellins synthesis and signal transduction, both known to be involved in fundamental plant physiological processes and correlated also to biotic stress response [46]. In the case of IAA, in V and MV plants, we observed the induction of several genes coding for auxin efflux carriers as well as several auxin-responsive factors (Appendix A and Appendix A). It is worth noting that for all these hormones, the highest number of DEGs related to their metabolism, signal transduction, and response was observed in V plants while the lowest number of related DEGs was found in M plants; an intermediate situation was observed in MV plants (Appendix A and Appendix A). In addition, an enrichment of DEGs related to cytokinins synthesis and degradation was found in MV plants only (Appendix A and Appendix A). In the case of ABA, JA, and SA, a DEGs enrichment was not clearly observed. However, we observed that, in the case of ABA, the gene coding for the (+)-abscisic acid 8’-hydroxylase, the enzyme involved in the oxidative degradation of ABA and known to play an important role in determining its level, was downregulated in V plants and not regulated in M and MV plants. On the other hand, in MV and particularly in M plants, we found induced the UDP-glucosyltransferases involved in the homeostasis of ABA. Finally, in V and MV plants, we observed the activation of few ABA-responsive genes. Considering JA, we found the JA-amido synthetase JAR1-coding genes that catalyze the synthesis of jasmonates-amino acid conjugates by adenylation slightly induced in V plants and slightly repressed in M and MV plants. The gene coding for the S-adenosyl-L-methionine carboxyl methyltransferase as well as those coding for methyl jasmonate esterase, all involved in the conversion from JA to Methyl-JA, were induced in V and MV plants but not in M plants. Finally, in the case of SA, in V plants, the phenylalanine ammonia-lyase (PAL) pathway showed the upregulation of a gene coding for a chorismate mutase 2 and one coding for a PAL, while a second PAL-encoding gene was downregulated; in the same plants, the chorismate pathway for the SA synthesis was downregulated. Moreover, we observed a downregulation of genes involved in the conversion from Methyl-SA to SA and the induction of a gene coding for UDP-glucose SA glucosyltransferase involved in the SA glycosylation. In M plants, we observed only a slight induction of one PAL-encoding gene and the induction of two genes involved in the conversion of SA to Methyl-SA. Finally, in MV plants, a slight downregulation of one PAL-encoding gene and one gene belonging to the chorismate pathway were observed. In these plants, two genes involved in the conversion of SA to Methyl-SA were also downregulated (Appendix A).

*Reactive oxygen species (ROS)*. In V plants only, we observed the induction of genes related to ROS synthesis (respiratory burst oxidases, RBOH, and peroxidases, POS) that suggests an increase in ROS production. In the same plants, the activation of glutathione-S-transferases (GST)-encoding genes and a general transcriptional perturbation of genes involved in ROS imbalance was highlighted (Appendix A). On the contrary, in MV plants, the downregulation of genes involved in ROS production, particularly POS, suggested that in these plants the production of ROS in response to viral infection was mitigated by mycorrhization (Appendix A). Moreover, in MV plants, genes involved in the ROS-detoxifying process mediated by the glutathione regenerating cycle, that is, monodehydroascorbate reductase (MDHAR), dehydroascorbate reductase (DHAR), glutathione reductase (GR), and catalase (CAT), were induced; in the same plants, GST and other enzymes involved in ROS imbalance such as alternative oxidases (AOX), methionine sulfoxide reductase (MSR), thioredoxins (TRX), and glutaredoxins (GRX) were also upregulated in respect to C plants. In M plants, the number of DEGs related to ROS production and imbalance was significantly lower than in V and MV plants, thus suggesting that, in these plants, these processes are not drastically affected (Appendix A).

*Other secondary metabolites.* In V plants, transcriptome analysis highlighted the differential expression of genes related to the biosynthesis of almost every pool of compounds involved in plant secondary metabolism (Appendix A). Similar to what was observed for stress responses and primary carbon metabolism, in MV plants the presence of AMF mitigated the impact of CMV by reducing both the up and downregulation of these genes. In particular, in both V and MV plants, we observed an overrepresentation of DEGs involved in the metabolism of waxes, cuticular complex mixtures of very long chain fatty acids able to form a natural barrier on aerial surfaces of terrestrial plants against biotic and abiotic stresses [47]. These genes were more downregulated in V than in MV plants and not differentially expressed in M plants (Appendix A).

## 4. Discussion

It is known that the modulation of plant defense responses may occur during mycorrhiza formation, potentially through hormone-dependent signaling pathways. This phenomenon, named mycorrhiza-induced resistance [27], may “prime” plants to deploy faster and/or stronger defense responses in the presence of an incoming stress, like a pathogen attack [25]. Few key criteria have been proposed to assess the effective presence of defense priming in a given biological system, that is, more robust defense responses, memory, better performance, and low fitness costs [25]. Evaluation of our results in the light of these criteria indicated that mycorrhization is able to prime tolerance against CMV infection in tomato plants.

Concerning the induction of more robust defense responses, we observed that the development of CMV infection was influenced by the AMF, that exerted an initial (14 dpi) protective role against viral symptoms development; however, this phenomenon disappeared over time, possibly as a consequence of a recovery occurred in the virus-infected plants [48]. The observed dynamic can be related to the strong silencing suppression activity of the viral protein 2b in the early stage of infection, which allows the virus to prevail on the plant defenses and develop strong symptoms. Then, in later stages, the coat protein of CMV may accumulate and induce a potent antiviral silencing state in the emerging tissues, which reduces the level of 2b silencing suppressor and allows the production and amplification of siRNAs that counteract viral replication [48]. Since viral concentration was similar in virus-infected plants regardless of AMF colonization, we infer that the mycorrhization may interfere with systemic symptoms development but not with viral replication. Indeed, in the case of CMV infection, symptoms recovery, in addition to the viral titer, may also be affected by the phosphorylation state of the viral coat protein [49]. Moreover, for other viruses, the viral titer is not always directly correlated with symptomatology [50,51]. Our results are consistent with those of [37] showing a reduction in viral symptoms in mycorrhizal tomato plants infected by tomato yellow leaf curl Sardinia virus (a DNA virus) when compared with those infected but not mycorrhized. However, our results are in contrast with those reported by other studies, performed on different RNA viruses, highlighting that the AMF colonization may lead to a worsening of symptoms and to an increase of viral titer over time [34,35,52,53]. Overall, these data suggest that the mycorrhization interferes with dynamics of viral infection and symptomatology in a way that likely depends on the specific combination of virus and plant species involved.

The SA accumulation observed in CMV-infected mycorrhizal plants is consistent with a mycorrhiza-induced priming effect, as SA has been shown to mediate priming against biotrophs during the challenge but not in the pre-challenge stage [54]. This SA accumulation was not mirrored by a modulation of genes belonging to the SA biosynthesis, suggesting the occurrence of post-transcriptional regulation events (Appendix A).

Beside SA, other hormones are known to be involved in the complex signaling network that modulates responses to abiotic and biotic stimuli [55,56] and play a role in priming plant defense [57]. ABA is well known to regulate stomata opening and is a key hormone in modulating plant responses to abiotic stresses [58]. Recently, the role of ABA in regulating both pathogenic and mutualistic interactions, and also the priming phenomenon, has become increasingly clear [57,59]. In our study, the increase of ABA level in M plants may be indicative of a pre-challenge primed state [60]. At the same time point, in MV plants (condition corresponding to the challenge state), ABA level showed a trend towards a decrease in favor of SA accumulation, possibly due to the antagonistic role of ABA and SA in defense responses [46]. This pattern seems to be in agreement with the activation of genes related to the ABA homeostasis in M and MV plants (Appendix A).

Particularly in M plants, we also observed an increase in the content of IAA, a hormone associated with the mycorrhizal development that may affect plant growth through influencing the symbiont activities [61,62]. The direct IAA measurement fits with the activation of the indole-3-acetic acid-amido synthetase GH3.8 coding gene, involved in the IAA synthesis, as well as of two auxin efflux carrier coding genes in M plants. In V and MV plants, the reduction of IAA in respect to control plants could be partially correlated to the transcriptional induction of several auxin efflux carriers involved in the IAA movement along the plant, but seems to be in contrast with the activation of several genes belonging to the IAA synthesis pathway in V plants (Appendix A). Overall, the involvement of the IAA in inducing a primed state, even if not to exclude it [63], needs further investigations.

More robust defense responses seem to be also indicated by the regulation of genes related to ROS production and imbalance highlighted by our transcriptomic analysis. ROS are usually generated in response to a wide variety of adverse environmental conditions including pathogen attack; they can play a dual role in plant cells, acting either as toxic molecules inducing programmed cell death or as signal transduction molecules [64,65]. As expected, in V plants, the response to infection was characterized by the induction of several genes related to ROS production and imbalance [66]. On the contrary, in MV plants, we specifically observed the activation of ROS-detoxifying process mediated by the glutathione regenerating cycle, in parallel with the repression of genes involved in ROS production. These results suggest that the mycorrhization can prevent the accumulation of ROS at toxic levels induced by viral infection.

It has been recently highlighted that when a plant is primed, the information of the priming stimulus is stored until exposure to a triggering stress; this process, called memory, implies the activation of several families of transcription factors, pattern recognition receptors and mitogen-activated protein kinases [25]. Interestingly, in our M plants (that correspond to the pre-challenge state), we have observed the induction of several WRKY, bHLH, bZIP, C2C2 CO-like, C2C2-Dof, MYB, and MYB-related transcription factors as well as mitogen-activated protein kinases (Appendix A). Our data are in agreement with what was observed in a previous RNA-seq study performed on leaves of tomato plants colonized by the AMF *Rhizophagus irregularis* [67], where several transcription factor families (WRKY, MYB, bHLH, bZIP and AP2/EREB) were represented among the mycorrhiza-responsive genes, possibly linked to the priming effect.

The positive effect exerted by the AMF on plant performances was clearly highlighted by the reduced impact of viral infection on the host photosynthetic process in the MV plants. Indeed, in V plants, a reduced stomatal conductance and a general downregulation of genes related to the photosystem II reaction center was evident, whereas, a reduction of these parameters was barely detected in MV plants. This was in agreement with the protective role of AMF already observed in response to both abiotic [68] and biotic stresses [69].

Concerning the low fitness cost aspect, we found that the presence of the AMF counterbalanced the observed decrease in the photosynthetic activity (evaluated through the instantaneous photosynthetic CO_2_ assimilation rate) in V in respect to C plants, restoring a pre-challenging state (M plants).

Overall, at the molecular level, CMV infection induced the most dramatic perturbation in tomato leaves transcriptome whereas the colonization by the AMF caused only a limited transcriptional response. The number of mycorrhiza-responsive genes in our tomato plants was similar to that previously reported for the tomato/*R. irregularis* interaction, where transcriptional changes regarded the functional categories proteins, RNA, signaling, transport, biotic/abiotic stresses, hormone metabolism, as well as priming of systemic defense [67]. Our results confirm that, even if the AMF is physically present only in the root system, it can exert a systemic effect on the aerial part of the plant [67,70]. As we observed here and previously with tomato spotted wilt virus [34], the colonization by an AMF exerts a systemic effect able to mitigate the host plant responses to viral infection. Moreover, we highlight that the transcriptional response of MV plants is not simply additive; indeed, 42% of DEGs were specific of this condition, thus indicating that the tripartite interaction is a complex system, not resulting only from the sum of single effects.

Among DEGs related to secondary metabolism, several genes involved in the formation of cuticular waxes were downregulated by CMV (V plants) but not by AMF (M plants), although the same genes were either slightly repressed or not regulated when CMV and AMF were simultaneously present (MV plants). It has been recently observed that CMV can cause a significant reduction in the polarization percentage of light reflected from the abaxial surfaces of leaves [71], probably through the alteration of cuticular waxes synthesis. This influences aphids colonization or feeding behavior, determining whether a plant is recognized as a suitable host. We can speculate that the AMF presence in CMV-infected plants could interfere on the complex relationship occurring between the virus and its vector, thus having an indirect effect on the viral spread. Further experiments will be necessary to better investigate this aspect.

In conclusion, in this study, we show that the mycorrhization is able to prime tomato plants, enhancing their tolerance against CMV infection, and that this seems not to be directly correlated with viral replication. In particular, we showed that *F. mosseae* colonization strongly mitigates the disruptive physiological and transcriptomic changes induced by viral infection and may affect higher trophic levels (i.e., insect vectors), thus underlining the complex ecological role of mycorrhizal fungi in the ecosystem.

## Figures and Tables

**Figure 1 viruses-12-00675-f001:**
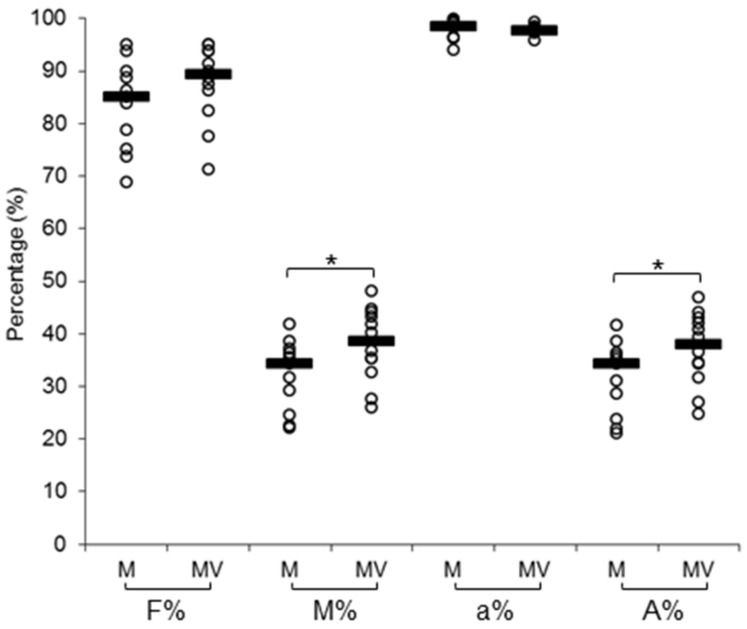
Mycorrhization levels of mycorrhizal (M) and mycorrhizal virus-inoculated (MV) tomato plants at 28 days post virus infection (= 56 days post inoculation with mycorrhizal inoculum). F%, mycorrhization frequency; M%, mycorrhization intensity; a%, arbuscules in the colonized segments of roots; A%, arbuscules in the whole root (n of samples = 12). Asterisks indicate significant differences (one-way ANOVA with post hoc Tukey, *p* < 0.05).

**Figure 2 viruses-12-00675-f002:**
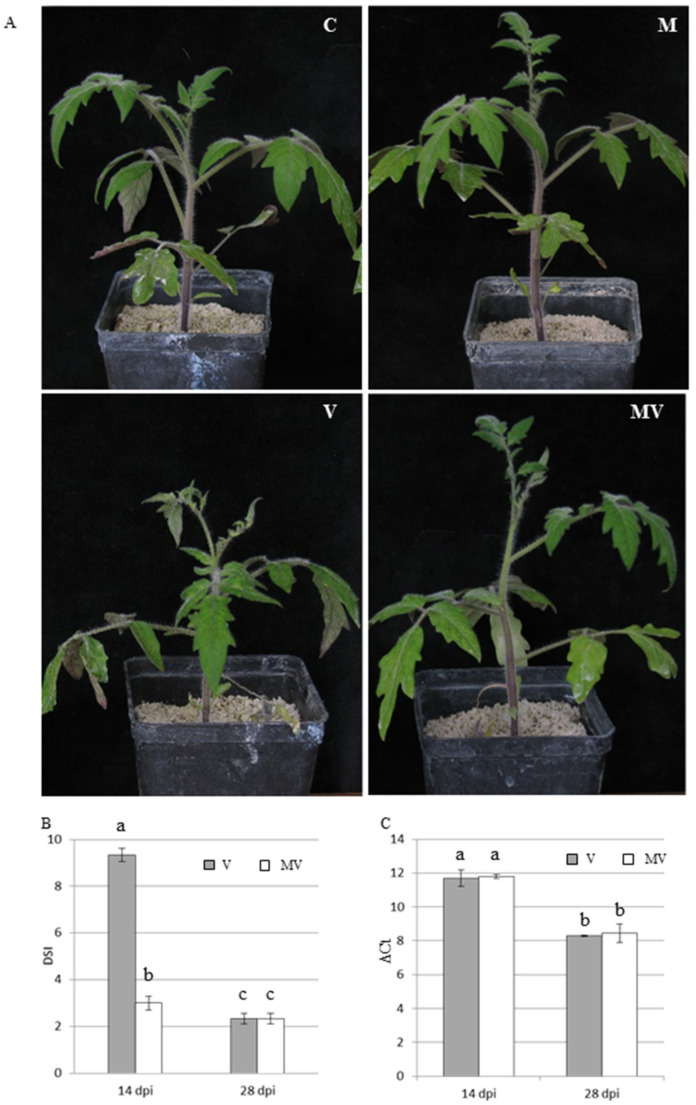
(**A**) Representative pictures of control (C), mycorrhizal (M), virus-inoculated (V), and mycorrhizal virus-inoculated (MV) tomato plants at 14 days post inoculation (dpi); (**B**) disease severity index (DSI) evaluated in V and MV tomato plants at 14 and 28 dpi; (**C**) quantification of viral replication by qRT-PCR in leaves of V and MV tomato plants at 14 and 28 dpi. ΔCt values on the vertical axis represent the difference between Ct (threshold cycle) of cucumber mosaic virus (CMV) and Ct of the reference gene *ubiquitin conjugating enzyme* (*UBC*), and estimate viral replication in V and MV plants. Vertical lines represent standard deviations. Different letters indicate statistically significant differences (Kruskal–Wallis test).

**Figure 3 viruses-12-00675-f003:**
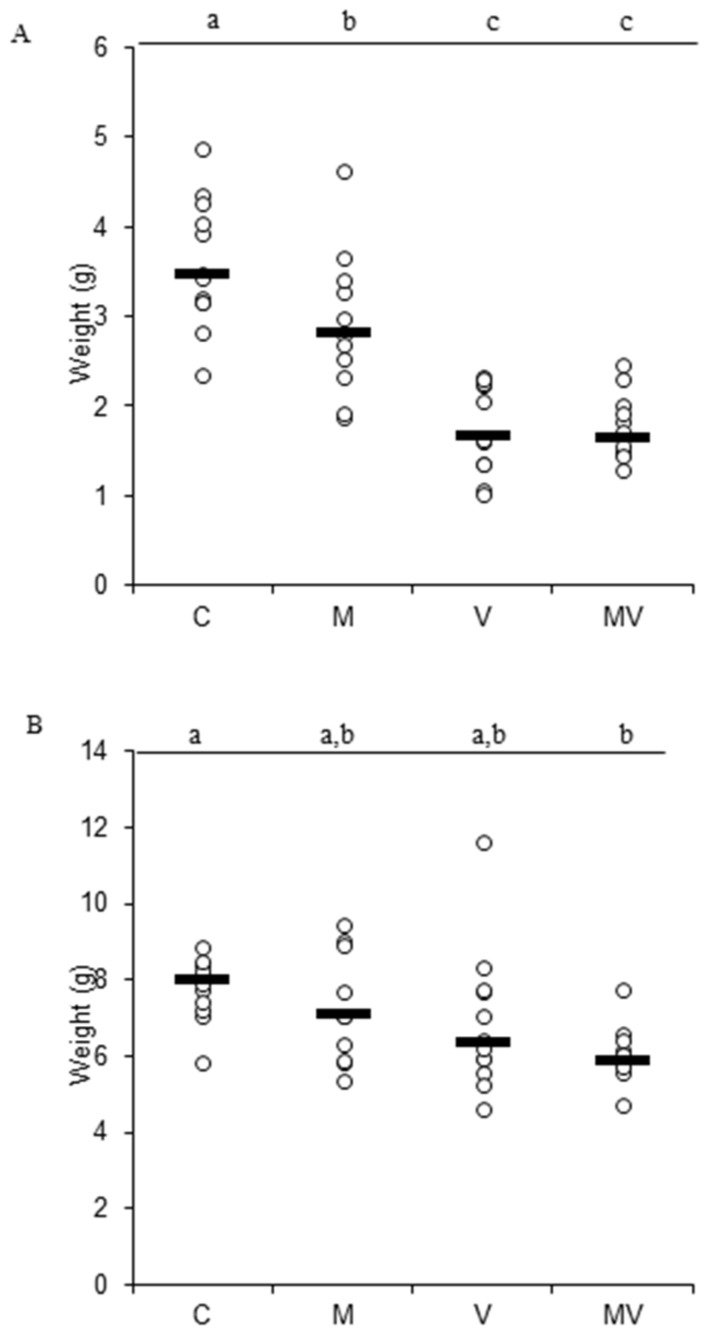
Hypogean (**A**) and epigean (**B**) biomass evaluation of control (C), mycorrhizal (M), virus-inoculated (V), and mycorrhizal virus-inoculated (MV) tomato plants at 28 days post inoculation (dpi) (n of samples = 12). Different letters indicate significant differences (one-way ANOVA with post hoc Tukey, *p* < 0.05).

**Figure 4 viruses-12-00675-f004:**
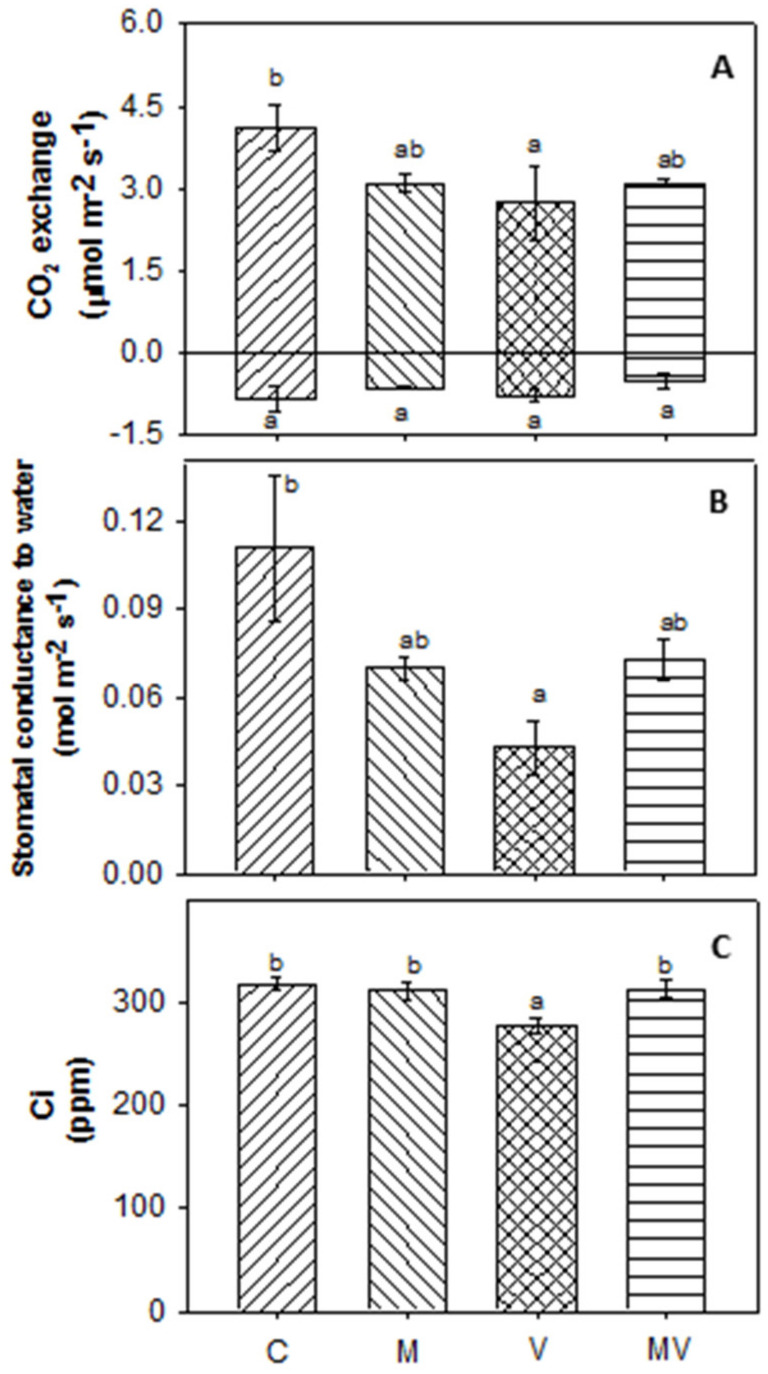
Evaluation of physiological parameters in mock-inoculated (C), mycorrhizal (M), virus-inoculated (V) and mycorrhizal virus-inoculated (MV) plants. (**A**, upper panel) photosynthetic CO_2_ assimilation rate, (**a**, lower panel) CO_2_ respiration rate, (**B**) stomatal conductance, (**C**) substomatal CO_2_ concentration (Ci). Data are means of five values (*n* = 5) ± standard deviation. Superscript letters indicate significant differences at *p* < 0.05 (Tukey’s test).

**Figure 5 viruses-12-00675-f005:**
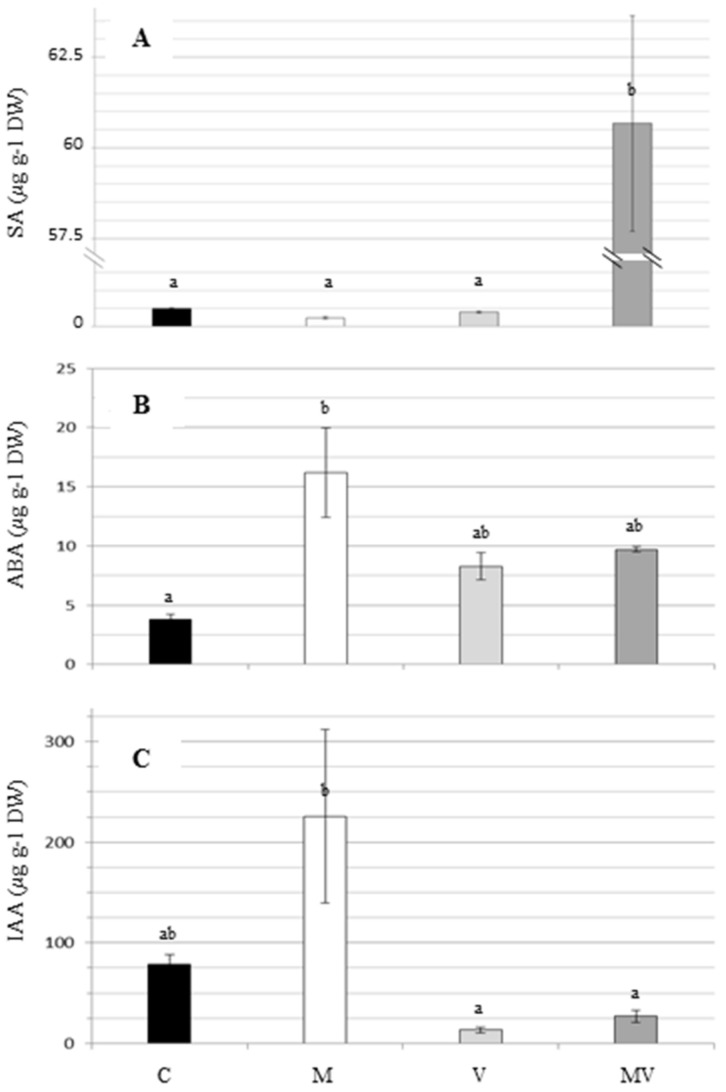
Evaluation of the levels of (**A**) salicylic acid (SA), (**B**) abscisic acid (ABA), and (**C**) indolacetic acid (IAA) in tomato leaves of mock-inoculated (C), mycorrhizal (M), virus-inoculated (V), and mycorrhizal virus-inoculated (MV) plants. Values are the average of three or four independent samples, each obtained by pooling four plants. Bars represent standard errors. Different letters indicate significant differences (one-way ANOVA with post hoc Tukey, *p* < 0.05). DW, dry weight.

**Figure 6 viruses-12-00675-f006:**
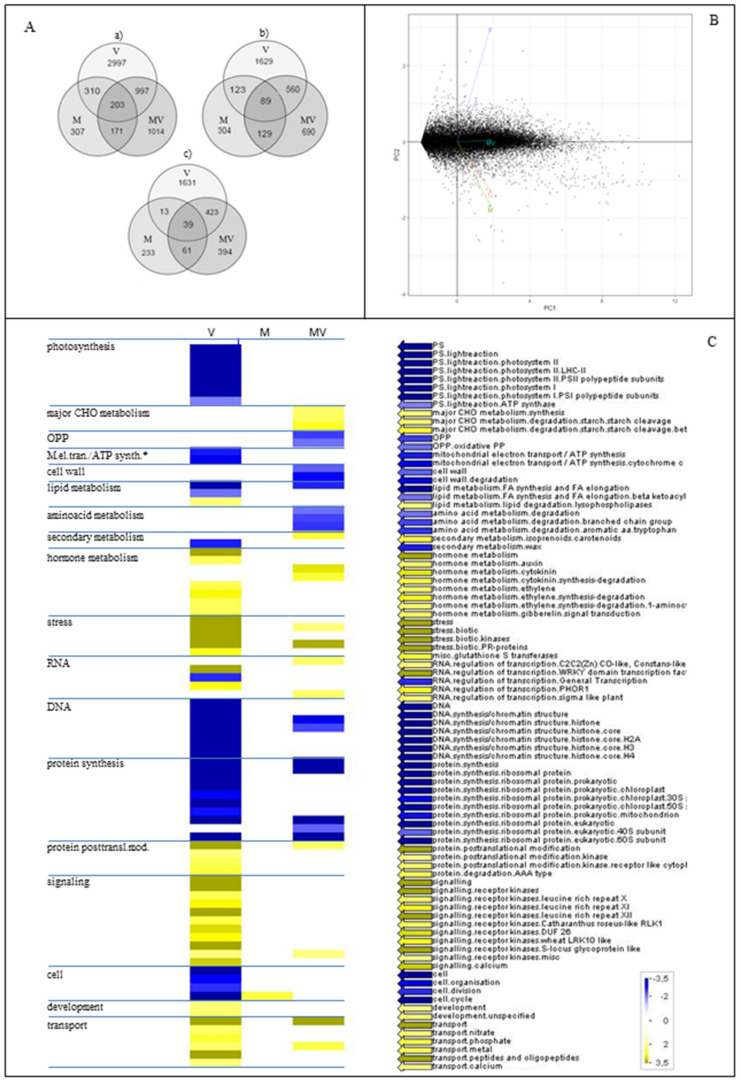
Transcriptomic data. (**A**) Number of (a) differentially expressed, (b) upregulated, and (c) downregulated genes in mycorrhizal (M plants), virus-inoculated (V plants), and mycorrhizal virus-inoculated (MV plants) tomato plants in respect to mock-inoculated plants (C plants). Numbers indicate differentially expressed genes; number of genes responsive in more than one condition is shown in the overlapping portion. (**B**) PCA analysis of the transcriptomic data. (**C**) Overview of MapMan functional categories overrepresented among DEGs. Expression values are reported as the log_2_ of fold change in respect to C plants; yellow and blue colors indicate up and downregulation, respectively, according to the legend reported in the figure.

## Data Availability

Raw fastq files were deposited in the Sequence Read Archive database of the National Center for Biotechnology Information under acc. num. SRP109092.

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
