# Peer review of "Arbuscular Mycorrhizal Symbiosis Primes Tolerance to Cucumber Mosaic Virus in Tomato"

_viruses, 2020, doi:10.3390/v12060675_

Round 1

Reviewer 1 Report

The manuscript by Miozzi et al., is well written and has new information worth publication. The authors, however, may consider the following editorial comments when revising the manuscript.

Major comments:

1-The manuscript contains many abbreviations which are hard to remember by the readers, and many of these abbreviations are inserted at different parts parts of the text. It is highly suggested to have a special section on abbreviations after keywords and before introduction, beginning of the introduction or at the end of the first page. The abbreviations may be created when a word or words are repeated in the text three times or more. Abbreviations may be divided into categories so it would be easy to spot. There is no need to explain in the text what an abbreviation means which happens most often in the present manuscript.

Supplementary tables (S Tables) need organization and accuracy. For example, files with names Table S5 to Table S10 are changed to Tables S1 to S6 when files are opened after clicking on file names. Thus the open file has a different table number than the name of the file which is confusing and inaccurate. As a result, there are duplicates of Tables S1 to S4 which differ in contents. Each open file of a table should have an accurate number of the table and a table title.

-Four columns in Table S5 which became S1 show numbers with dots such as 10.552.060 for total reads, these dots should be changed to commas to read 10,552,060. This correction should be applied to all numbers in these four columns.

-No data were presented in Table S8 which became S4. Either delete the table or show the missing data.

Other Comments:

1-Lines 4-5: The number after the name of each author should be in superscript. The same for the asterisk after the name of two authors.

2-Line 25: Bromoviridae should be in italic as recommended by ICTV

3-Line 26: Funneliformis mosseae should be in italic

4-Line 26: When dealing with virus taxonomy, the virus species should be written in italic according to ICTV. Accordingly, change "CMV, species Cucumber mosaic cucumovirus" to: CMV, species Cucumber mosaic virus. For your information the genus name of the virus is Cucumovirus which is written in italic.

5-Line 31: CO2: 2should be subscript

6-Line 32: What does the abbreviation ROS stand for. Provide the information in the abstract.

7-Line 36: Change Cucumber mosaic cucumovirus to cucumber mosaic virus

8-Lines 37-38: Solanum lycopersicum, the scientific name for tomato should be in italic. The same applies for any plant scientific name throughout the manuscript.

9-Line46: Revise (insect/fungi) to: (insect/fungi and others)

10-Line 78: See comment of line 25

11-Line 79: -----in tomato. Write the scientific name of tomato in italic here as this is the first time tomato mentioned in the text. Delete the scientific name of tomato from line 81 as it is not needed

12-Line 86: S. lycopersicum should be in italic

13-Lines 91-92: The authors are repeating what was written before by stating: control plants (C), CMV-infected (V), erc. Either write the abbreviations only OR delete the abbreviations in the two lines

14-Line 97: Move (syn. Glomus mosseae) Gerd & Trappe to line 78 next to F. mosseae. Both scientific names should be in italic

15-Line 97: What is meant by BEG12? Clarify

16-Line 105: Nicotiana benthamiana should be initalic

17-Lines 110-111: See comment of line 105

18-Line 171: CO2 and H2O: 2 should be subscript

19-Lines 174-177 and 180: CO2: 2 should be subscript

20-Line 196: Change 15'000 to: 15,000

21-Lines 210-212: Is the sentence that starts on line 210 and ends at the beginning of line 212 needed?  This is a repeat and it is not part of the results

22-Lines 223-224 and 268: F. Mosseae should be in italic

23- Figure 3: Present V first then MV second for both upper and lower panels and revise the figure legend accordingly (See Figure 4).

24-Lines 265-266, 270-271 and 274: CO2: the number should be subscript

25-Line 396: Delete the abbreviation TYLCSV and replace it by the virus name: tomato yellow----

26- Line 412: Topoviridae should be in italic

27-Line 426: Revise "associated to" to: associated with

28-Line 438: Revise" to toxic levels" to: at toxic levels

29- Line 459: CO2: 2 should be subscript

30-Lines 463-464: "resulted similar to" delete resulted and replace it with was to read: was similar to

31-Line 478: Change Mauck and coauthers to: Mauck et al.,

32-Line 478: Insert insect before non-persistently

33-Line 480: Change Volpe and coauthors to Volpe et al.,

34-Line 481: Write the complete scientific name of R. irrigularis in italic

35-Lines 481-483: Aphid scientific names should be in italic

36- Line 500: F. mosseae should be in italic

37-Reference: The title of some references is not written according to the journal guidelines. That should be corrected

Author Response

1-The manuscript contains many abbreviations which are hard to remember by the readers, and many of these abbreviations are inserted at different parts of the text. It is highly suggested to have a special section on abbreviations after keywords and before introduction, beginning of the introduction or at the end of the first page. The abbreviations may be created when a word or words are repeated in the text three times or more. Abbreviations may be divided into categories so it would be easy to spot. There is no need to explain in the text what an abbreviation means which happens most often in the present manuscript.

We have added a list of abbreviations just before the Introduction; the list is divided in categories as requested. For the abbreviations in the text we have followed these rules: abbreviations used in the abstracts are reported in parentheses the first time and then used in the abstract itself. In the text, the abbreviations are reported in parentheses the first time and then used in the whole manuscript. Definition of abbreviations is also reported in the figure legends in order to facilitate their interpretation. To facilitate the reading, we avoid to insert in the list, the abbreviation of  genes named only once in the text.

Supplementary tables (S Tables) need organization and accuracy. For example, files with names Table S5 to Table S10 are changed to Tables S1 to S6 when files are opened after clicking on file names. Thus the open file has a different table number than the name of the file which is confusing and inaccurate. As a result, there are duplicates of Tables S1 to S4 which differ in contents. Each open file of a table should have an accurate number of the table and a table title.

-Four columns in Table S5 which became S1 show numbers with dots such as 10.552.060 for total reads, these dots should be changed to commas to read 10,552,060. This correction should be applied to all numbers in these four columns.

-No data was presented in Table S8 which became S4. Either delete the table or show the missing data.

All problems with the Tables in the supplementary files have been fixed. Table S8 (now S10) is not empty but it is composed of several excel sheets. The first one reports the title of the table and the others the data.

We apologize for the several format errors in the file but they were not present in the submitted world file so we guess there was a problem at the submitting step. Anyway, we have fixed all the following points.

Other Comments:

1-Lines 4-5: The number after the name of each author should be in superscript. The same for the asterisk after the name of two authors.

2-Line 25: Bromoviridae should be in italic as recommended by ICTV

3-Line 26: Funneliformis mosseae should be in italic

4-Line 26: When dealing with virus taxonomy, the virus species should be written in italic according to ICTV. Accordingly, change "CMV, species Cucumber mosaic cucumovirus" to: CMV, species Cucumber mosaic virus. For your information the genus name of the virus is Cucumovirus which is written in italic.

5-Line 31: CO2: 2should be subscript

6-Line 32: What does the abbreviation ROS stand for. Provide the information in the abstract.

7-Line 36: Change Cucumber mosaic cucumovirus to cucumber mosaic virus

8-Lines 37-38: Solanum lycopersicum, the scientific name for tomato should be in italic. The same applies for any plant scientific name throughout the manuscript.

9-Line46: Revise (insect/fungi) to: (insect/fungi and others)

10-Line 78: See comment of line 25

11-Line 79: -----in tomato. Write the scientific name of tomato in italic here as this is the first time tomato mentioned in the text. Delete the scientific name of tomato from line 81 as it is not needed

12-Line 86: S. lycopersicum should be in italic

13-Lines 91-92: The authors are repeating what was written before by stating: control plants (C), CMV-infected (V), erc. Either write the abbreviations only OR delete the abbreviations in the two lines

14-Line 97: Move (syn. Glomus mosseae) Gerd & Trappe to line 78 next to F. mosseae. Both scientific names should be in italic

15-Line 97: What is meant by BEG12? Clarify

Banque Européenne de Glomales (BEG) now International Bank for the Glomeromycota is the fungal collection where this fungal isolate is deposited (https://www.i-beg.eu/cultures/BEG12.htm).

The text has been changed into “isolate BEG12”

16-Line 105: Nicotiana benthamiana should be in italic

17-Lines 110-111: See comment of line 105

18-Line 171: CO2 and H2O: 2 should be subscript

19-Lines 174-177 and 180: CO2: 2 should be subscript

20-Line 196: Change 15'000 to: 15,000

21-Lines 210-212: Is the sentence that starts on line 210 and ends at the beginning of line 212 needed?  This is a repeat and it is not part of the results

22-Lines 223-224 and 268: F. Mosseae should be in italic

23- Figure 3: Present V first then MV second for both upper and lower panels and revise the figure legend accordingly (See Figure 4).

24-Lines 265-266, 270-271 and 274: CO2: the number should be subscript

25-Line 396: Delete the abbreviation TYLCSV and replace it by the virus name: tomato yellow----

26- Line 412: Topoviridae should be in italic

27-Line 426: Revise "associated to" to: associated with

28-Line 438: Revise" to toxic levels" to: at toxic levels

29- Line 459: CO2: 2 should be subscript

30-Lines 463-464: "resulted similar to" delete resulted and replace it with was to read: was similar to

31-Line 478: Change Mauck and coauthers to: Mauck et al.,

32-Line 478: Insert insect before non-persistently

33-Line 480: Change Volpe and coauthors to Volpe et al.,

34-Line 481: Write the complete scientific name of R. irrigularis in italic

35-Lines 481-483: Aphid scientific names should be in italic

36- Line 500: F. mosseae should be in italic

37-Reference: The title of some references is not written according to the journal guidelines. That should be corrected

All these points have been corrected

Reviewer 2 Report

The authors investigate the effects of virus infection on tomatoes that form AM mycorrhiza, and have stated that AM colonization would induce (prime) resistance to the virus (cucumber mosaic virus). How the presence of AM colonization influences viral infection (reduces or exacerbates viral symptoms) is thought to vary depending on plant and virus combination, and thecomprehensive explanation is not well obtained.

The paper reports that AM colonization reduces the CMV symptoms. However, at 14 dpiays, the viral accumulation was the same with or without mycorrhiza. Also, at 28 dpi, the difference in symptoms has disappeared. The problem is that there is no convincing explanation for the lack of correlation between virus accumulation and disease symptoms (i.e.,different disease symptoms with no difference in virus titer). It is an important viewpoint as an academic journal of virology.

In addition, is it reasonable to conduct experiments under the condition that mycorrhiza formation does not affect viral replication? ? In addition, AM colonization did not increase tomato biomass with or without viral infection. With such experimental materials, I think that the relationship between mycorrhiza formation and virus infection cannot be analyzed from a biologically meaningful viewpoint. Therefore, if you want to show that AM colonization induces viral resistance, I propose to use materials and methods that can more reliably prove it.

On the other hand, the transcriptome results (Fig. 6) and the hormonal analysis results (Fig. 5) are interesting because they show the results specific to each test group. However, these data and the stories from Fig. 1 to Fig. 4 are hardt to be connected, so it would difficult to show as a single story.

Therefore, it is judged that this paper has not reached the quality to be published as an academic paper, and I would like to request a major revision of the paper composition.

Other points:

p6, line 236, viral load -> viral titer or viral accumulation is better

p8, line 270, Fig. 4C? (not Fig. 4D)

p9, line 277, Fig. 4D? (not Fig. 4C)

p10, line 313, Permanent Component Analysis?? (is this Principal Component Analysis?)

Fig.1 M% (mycorrhizal intensity), This word is not well understood for people who are not in the same research field. Please provide an explanation that will help our understand.

Fig.2 There are no difference in viral symptoms at 28 dpi, but did author check whether symptoms reappear in plants without AM infection, or not? ? In addition, is there a difference in virus accumulation before 14 dpi?

Fig.6 The contents are not clearly visible because the characters are unclear. This figure should be replaced with a better resolution.

Author Response

The authors investigate the effects of virus infection on tomatoes that form AM mycorrhiza, and have stated that AM colonization would induce (prime) resistance to the virus (cucumber mosaic virus). How the presence of AM colonization influences viral infection (reduces or exacerbates viral symptoms) is thought to vary depending on plant and virus combination, and the comprehensive explanation is not well obtained.

The paper reports that AM colonization reduces the CMV symptoms. However, at 14 dpi, the viral accumulation was the same with or without mycorrhiza. Also, at 28 dpi, the difference in symptoms has disappeared. The problem is that there is no convincing explanation for the lack of correlation between virus accumulation and disease symptoms (i.e.,different disease symptoms with no difference in virus titer). It is an important viewpoint as an academic journal of virology.

We agree with the referee that this is an important point, so we added some comments and three references in the discussion section.

In addition, is it reasonable to conduct experiments under the condition that mycorrhiza formation does not affect viral replication? ?

We would like to underline that the aim of the present work was indeed to investigate whether the AM symbiosis could influence the viral infection (symptoms and viral replication), since little is known about the effect of mycorrhization on viral infection and available data are contradictory (see Miozzi et al., 2019). Indeed, nothing was known on the capability of the AM symbiosis to affect CMV replication before this study. 

In addition, AM colonization did not increase tomato biomass with or without viral infection. With such experimental materials, I think that the relationship between mycorrhiza formation and virus infection cannot be analyzed from a biologically meaningful viewpoint. Therefore, if you want to show that AM colonization induces viral resistance, I propose to use materials and methods that can more reliably prove it.

As explained in the Materials & Methods section, the plants were watered twice a week: once with a modified Long Ashton nutrient solution containing 320 μM phosphate (Pi) and once with water. Pi content was optimized in the nutrient solution so that control plants did not suffer from Pi limitations and have a biomass similar to that of mycorrhizal plants, as described in a previous publication Maffei et al. (2014). We set up these Pi conditions on purpose, in order to eliminate the effect of the AM symbiosis on the viral infection only mediated by the nutritional benefit and to highlight other processes. Under these conditions we do not expect a higher biomass in mycorrhizal vs non mycorrhizal plants, as we observe on the contrary when plants are grown at very low (3 μM) Pi.

On the other hand, the transcriptome results (Fig. 6) and the hormonal analysis results (Fig. 5) are interesting because they show the results specific to each test group. However, these data and the stories from Fig. 1 to Fig. 4 are hard to be connected, so it would be difficult to show as a single story.

Figures from 1 to 4 are important since they aim to characterize the biological conditions from different aspects and are propaedeutic to the following analyses shown in Fig. 5 and 6: in particular, Fig. 1 confirms that all the considered plants (M and MV) were successfully colonized by the arbuscular mycorrhizal fungus, thus confirming that they are comparable; Fig. 2 aims to show the effect of mycorrhization on viral infection, in terms of symptoms and viral titer; Fig. 3 characterizes the biomass in the considered conditions and Fig. 4 investigate physiological parameters mainly highlighting that viral infection in the absence of mycorrhization has the major impact on the plant; mycorrhization in MV plants seems to have a buffer effect on different physiological parameters, restoring a situation similar to M plants.   

Therefore, it is judged that this paper has not reached the quality to be published as an academic paper, and I would like to request a major revision of the paper composition.

Other points:

All the following points have been addressed

p6, line 236, viral load -> viral titer or viral accumulation is better

p8, line 270, Fig. 4C? (not Fig. 4D)

p9, line 277, Fig. 4D? (not Fig. 4C)

p10, line 313, Permanent Component Analysis?? (is this Principal Component Analysis?)

Fig.1 M% (mycorrhizal intensity), This word is not well understood for people who are not in the same research field. Please provide an explanation that will help our understand.

M% corresponds to the intensity of the mycorrhizal colonization within each root fragment analysed.

F%: corresponds to the number of colonized root fragments among those analysed

a%: corresponds to the arbuscule abundance in colonized fragments of the root

A%: corresponds to the arbuscule abundance in the whole root system

These definitions have been introduced in the Materials and Methods section.

Fig.2 There are no difference in viral symptoms at 28 dpi, but did author check whether symptoms reappear in plants without AM infection, or not? ? In addition, is there a difference in virus accumulation before 14 dpi?

At 28 dpi, all the plants were sacrificed in order to verify the level of mycorrhization and evaluate the biomass of the epigean and hypogean parts, therefore we were not able to check symptoms after this time point. We didn’t check virus accumulation before 14 dpi; this time point was chosen as the first time point where the viral symptoms were clearly visible in V plants. A part related to viral titre and symptomatology has been added in the discussion.

Fig.6 The contents are not clearly visible because the characters are unclear. This figure should be replaced with a better resolution.

 Fig.6 has been changed with a figure with a higher resolution

Reviewer 3 Report

Miozzi, Vaira and co-workers report interesting the result of the development of CMV infection in tomato plants and the exertion of AMF priming effect that enables an enhanced tolerance to CMV infection at physiological, biochemical and molecular level. It was shown that arbuscular mycorrhizal fungi could be used for as a biocontrol of CMV. In this study, the authors show the mechanisms involved in the mycorrhiza-induced tolerance to CMV infection based mainly on the transcriptomic data. However, direct evidence needs to be shown to prove these mechanisms in the control of CMV infection. Specific comments are as follows. It is necessary to rewrite the text based on the new results.

  1. Overall, this manuscript seems to have limited specialized content about CMV. Despite the fact that the target of control is CMV, a detailed description of CMV is missing in the introduction, so a description is required.

  1. The abstract or Page. 6 line 237-238 state“the establishment of a functional AM symbiosis delays symptoms development without interfering with CMV replication.” or “ the mycorrhizal colonization influenced and delayed symptom development, but not virus accumulation.”. In plants infected with CMV, CMV  shows cyclic symptoms. In addition, from the CMV research field standpoint, it is also important to discuss about the effects of RNA silencing when considering plant-virus interactions. Therefore, drawing conclusions from the only data in Figure 2B and 2C could be led to misunderstanding. Authors should add data to prove it  and please rewrite the text clearly to avoid confusion.

  1. Photographs of the symptoms of tomatoes are presented in Figure 2, however the pictures showing CMV-infected tomatoes do not illustrate the symptoms clearly. Clearer photographs are required. It is recommended that the photographs depicting infected leaves be enlarged.

  1. CMV spread to the whole plants after multiplying in the roots. The results in Fig.3 need to be reconsidered from this point. Dry matter should be used to provide more accurate biomass data.

  1. Figure 5 shows the measurement of salicylic acid. Normally, CMV-infected plants detect an increase in salicylic acid, so the value of n.d. is incorrect. In fact, the transcriptomic data shows that the expression of shikimate pathway is elevated. This suggest that SA-synthesized enzymes are elevated in CMV-inoculated plants. The results appear inconsistent. In page 13 line 407-408, authors state “it is not likely to be significantly induced by CMV”. There have been reported that SA was detected in tomato plants infected CMV. Remeasurement is required. In addition, to avoid confusion, data of SA related genes should be added to Figure S4.

  1. In page 9, line 296, authors state “we measured the JA level in our plants but it was under the detection limit in all the conditions.”. To avoid confusion, data of SA related genes and JA related genes should be added to Figure S4.

  1. In Figure 7, authors show schematics of the mechanisms involved in the mycorrhiza-induced tolerance to CMV infection based on their transcriptomic data. Transcriptomic data alone is insufficient to determine these mechanisms. As such, more direct evidence is needed to prove these mechanisms. Page 15 , line 501-502 states “by increasing SA levels and avoiding excessive accumulation of ROS(Fig.7)”. At a minimum, ROS measurement is required.

Author Response

Miozzi, Vaira and co-workers report interesting the result of the development of CMV infection in tomato plants and the exertion of AMF priming effect that enables an enhanced tolerance to CMV infection at physiological, biochemical and molecular level. It was shown that arbuscular mycorrhizal fungi could be used as a biocontrol of CMV. In this study, the authors show the mechanisms involved in the mycorrhiza-induced tolerance to CMV infection based mainly on the transcriptomic data. However, direct evidence needs to be shown to prove these mechanisms in the control of CMV infection. Specific comments are as follows. It is necessary to rewrite the text based on the new results.

The presented work aims to expand the case studies on plant-virus-mycorrhiza interaction, extending our knowledge to a not yet investigated viral family (i.e. Bromvirideae); we performed a series of transcriptomic, physiological and hormonal analyses in order to investigate and better characterize the observed response of mycorrhizal plants to viral infection. We are aware that other studies are needed to precisely define the underlying mechanism, but this is beyond the aim of this work. We therefore rewrite accordingly parts of the text.

Overall, this manuscript seems to have limited specialized content about CMV. Despite the fact that the target of control is CMV, a detailed description of CMV is missing in the introduction, so a description is required.

We have added a paragraph dedicated to CMV description in the Introduction and we have added a reference that further describes the virus and its viral genes: MOCHIZUKI, T. and OHKI, S.T. (2012), Cucumber mosaic virus: viral genes as virulence determinants. Molecular Plant Pathology, 13: 217-225. doi:10.1111/j.1364-3703.2011.00749.x

The abstract or Page. 6 line 237-238 states “the establishment of a functional AM symbiosis delays symptoms development without interfering with CMV replication.” or “ the mycorrhizal colonization influenced and delayed symptom development, but not virus accumulation.”. In plants infected with CMV, CMV  shows cyclic symptoms. 

We prefer to use the term “recovery”, and not “cyclic symptoms'' for plants of our experiments. Cyclic symptoms are described in the literature for CMV infection on tobacco (see Sunpapao et al., 2011) as: “Following CMV infection, the uninoculated upper leaves of tobacco plants are alternately mosaic and mottled, then symptomless, and finally again became mosaic.” In our case we did not observe a second round of severe symptoms after recovery: the reason can be that we did not keep the tomato plants for longer (we had to measure the biomass, so we sacrificed them at 28 dpi). Or it is also possible that the reaction of tomato is not the same as tobacco. Anyway it’s worth noting that  the symptoms observed in V and MV plants at 14 dpi were strikingly different in spite of the analogous viral challenge treatment.

In addition, from the CMV research field standpoint, it is also important to discuss the effects of RNA silencing when considering plant-virus interactions. Therefore, drawing conclusions from the only data in Figure 2B and 2C could be led to misunderstanding. Authors should add data to prove it  and please rewrite the text clearly to avoid confusion.

We thank Reviewer #3 for making us aware that the sentence: “the establishment of a functional AM symbiosis delays symptoms development without interfering with CMV replication” is not correct. Reconsidering our data about symptoms and virus titre at the two timepoints considered, we see that in MV the symptoms remain always mild, they are therefore not delayed. On the other hand, in V plants, symptoms are severe at 14 dpi, and become mild at 28 dpi. What we see is a combination and overlap of two dynamic events: the spontaneous recovery of CMV infection (even without AMF) and the priming effect of the AMF. At 14 dpi the priming due to AMF cannot stop high CMV replication, but can efficiently compensate the phenotype (symptoms), avoiding a severe diseased condition of the plant. Later, at 28 dpi, the priming effect is not phenotypically visible any more, due to spontaneous recovery, although it remains strong in the plant, as the transcriptomic data (and the physiological parameters) show. The reduction in virus titre before and after recovery is actually interesting (Fig.2C).

This remarkable virus reduction merits a few more words: here we thank again Reviewer #3 for pointing our attention to the involvement of RNA silencing in what we observed. CMV carries a gene encoding a strong silencing suppressor, the 2b protein. The mechanisms that 2b protein activates to block the plant machinery from destroying the viral RNA has been deeply studied (Duan, C.-G., Fang, Y.-Y., Zhou, B.-J., Zhao, J.-H., Hou, W.-N., Zhu, H., et al. 2012. Suppression of Arabidopsis ARGONAUTE1-mediated slicing, transgene-induced RNA silencing, and DNA methylation by distinct domains of the Cucumber mosaic virus 2b protein. Plant Cell 24, 259–274. Doi: 10.1105/tpc.111.092718). When plants recover, the RNA silencing machinery is active, and can strongly reduce virus accumulation (and disease symptoms). The explanation of this dynamic change has been proposed in a recent paper (Zhang et al., 2017, PLoS Pathogens), where a role for the CMV CP has been demonstrated. While in an early stage of infection CMV, thanks to the strong silencing suppression activity of 2b protein, prevails over plant defences and invades the shoot apical meristems, causing growth of severely symptomatic leaves, in later stages the CMV CP accumulates and induces a potent antiviral silencing state in the emerging tissues, which reduces the level of 2b silencing suppressor and allows production and amplification of siRNA. The result is an attenuation leading to a compatible interaction between plant and virus. We modified the discussion introducing these aspects.

Photographs of the symptoms of tomatoes are presented in Figure 2, however the pictures showing CMV-infected tomatoes do not illustrate the symptoms clearly. Clearer photographs are required. It is recommended that the photographs depicting infected leaves be enlarged.

We agree with the referee's comment; for this reason we decided to modify Fig. 2 and show in Figure 2A only representative pictures of V and MV tomato plants, for which DSI and virus replication evaluation are shown in Figures 2B and 2C respectively. In this way the two pictures are larger and more detailed. In particular, the pronounced dwarfness of the plant and the curly/deformed appearance of young leaves in V in respect to MV can be better appreciated.

CMV spread to the whole plants after multiplying in the roots. The results in Fig.3 need to be reconsidered from this point.

We added a comment in the text, result section, from line 254

Dry matter should be used to provide more accurate biomass data.

We agree with the referee but we decided to not use dry matter for biomass evaluation in order to keep good material for transcriptomic and hormonal analyses

Figure 5 shows the measurement of salicylic acid. Normally, CMV-infected plants detect an increase in salicylic acid, so the value of n.d. is incorrect. In fact, the transcriptomic data shows that the expression of the shikimate pathway is elevated. This suggests that SA-synthesized enzymes are elevated in CMV-inoculated plants. The results appear inconsistent. In page 13 line 407-408, authors state “it is not likely to be significantly induced by CMV”. There have been reports that SA was detected in tomato plants infected with CMV. Remeasurement is required. In addition, to avoid confusion, data of SA related genes should be added to Figure S4.

We thank the reviewer for the valuable comments. Unfortunately no more material is available for repeating the analysis, so we carefully checked our measurements and we believe that they are correct. We recognize that probably we didn’t make clear enough that the HPLC-DAD measurements done for all the conditions, with the only exception of MV plants, were under the detection limit; therefore, in the conditions C, M and V we were unable to detect differences among the samples. On the other hand, in the MV condition, we could detect a certain amount of SA and we considered that finding as an indication of a further increase of SA in MV plants. The text has been changed to make clear this issue.

We would also like to point out that different host species may use significantly different mechanisms to cope with the infection by the same virus, as exemplified by Maters et al., 2005, and that, for instance, SA-induction after CMV infection has been observed in arabidopsis and tobacco (Zhou et al., 2014) but not in tomato (Vitti et al., 2016).

Following the suggestions of the referee, we focused our attention also on the shikimate pathway. As reported now in the TableS8 (sheet: SA-related genes) and figureS4, we found a down-regulation of a few enzymes of this pathway in V plants, thus supporting our SA measure.

In page 9, line 296, authors state “we measured the JA level in our plants but it was under the detection limit in all the conditions.”. To avoid confusion, data of SA related genes and JA related genes should be added to Figure S4.

According to referee indications, we added the data in Figure S4

In Figure 7, authors show schematics of the mechanisms involved in the mycorrhiza-induced tolerance to CMV infection based on their transcriptomic data. Transcriptomic data alone is insufficient to determine these mechanisms. As such, more direct evidence is needed to prove these mechanisms. Page 15 , line 501-502 states “by increasing SA levels and avoiding excessive accumulation of ROS(Fig.7)”. At a minimum, ROS measurement is required.

In the intention of the authors, Figure 7 was supposed to be a schematic proposal of a model for the tripartite tomato/CMV/F. mossae interaction, based on the integration of transcriptomic data as well as physiological and hormonal measurements. However, we may agree with the referee that direct evidence would be useful before proposing such a model. Therefore, we have removed Figure 7 and modified the discussion accordingly.

Round 2

Reviewer 2 Report

I think that authors responded well to the comments I made. I was worried that AM colonization did not have a positive effect on tomato biomass, but I understood that authors wanted to see the effect of virus infection on mycorrhizal plants under conditions that excluded such AM’s beneficial effect. However, it still gives the impression that there is insufficient consideration (discussion) of the results obtained. It should be revised so that a lot of results are summarized in an easy-to-understand manner and appropriate discussion is given.

Major points:

The result in Fig. 2 has been modified to show only V and MV plants, but pictures of C and V are also needed.

P12, line 367, In this manuscript, the word ‘symbiosis’ should not be used because AM infection does not show a growth promoting effect on tomatoes. AM infection or mycorrhization is recommended. Other parts are the same (P13, line 431, line 440, P15, line 501)

P12, line 367-369, In agreement with what observed ….. (Fig. 4A). This sentence is inappropriate. Fig 4A does not show such result; reduction of the photosynthetic performance by CMV infection was attenuated in MV plants.

P12, line 374-387 description about phytohormones in RNAseq results. It is not good that the contents described here and the results of the hormone quantification shown in Fig. 5 are not related. It is okay to include details about ethylene, but the gene expression associated with Fig.5 results should also be mentioned. In the discussion, it should be mentioned whether hormone quantification and transcriptome data were related or not.

I think the VOC results are unnecessary. No difference was found in the results for V, M, and MV plants, and the discussion part (p15, line 538-550) did not match the story of the paper.

Minor points,

The usage of ‘evolution’ in the sentence is not good

P7, line 268, by evaluating

Author Response

I think that the authors responded well to the comments I made. I was worried that AM colonization did not have a positive effect on tomato biomass, but I understood that authors wanted to see the effect of virus infection on mycorrhizal plants under conditions that excluded such AM’s beneficial effect. However, it still gives the impression that there is insufficient consideration (discussion) of the results obtained. It should be revised so that a lot of results are summarized in an easy-to-understand manner and appropriate discussion is given.

Major points:

The result in Fig. 2 has been modified to show only V and MV plants, but pictures of C and V are also needed.

Fig.2 has been modified

P12, line 367, In this manuscript, the word ‘symbiosis’ should not be used because AM infection does not show a growth promoting effect on tomatoes. AM infection or mycorrhization is recommended. Other parts are the same (P13, line 431, line 440, P15, line 501)

We changed the word ‘symbiosis’ with mycorrhization or AMF colonization

P12, line 367-369, In agreement with what observed ….. (Fig. 4A). This sentence is inappropriate. Fig 4A does not show such result; reduction of the photosynthetic performance by CMV infection was attenuated in MV plants.

 We corrected the sentence

P12, line 374-387 description about phytohormones in RNAseq results. It is not good that the contents described here and the results of the hormone quantification shown in Fig. 5 are not related. It is okay to include details about ethylene, but the gene expression associated with Fig.5 results should also be mentioned. In the discussion, it should be mentioned whether hormone quantification and transcriptome data were related or not.

RNAseq data on phytohormones have been added in the result section and the discussion has been modified accordingly

I think the VOC results are unnecessary. No difference was found in the results for V, M, and MV plants, and the discussion part (p15, line 538-550) did not match the story of the paper.

Following the referee’s advice we removed the VOCs part from the MS.

Minor points,

The usage of ‘evolution’ in the sentence is not good

Changed with ‘development’

P7, line 268, by evaluating

Corrected

Reviewer 3 Report

In this resubmitted manuscript, the authors changed the results (Abstract: line 29, "without interfering with CMV replication"), and modified the manuscript, and the authors additional experimental analysis of the dry data was performed. However, due to shortage of the stored samples, the wet data of required experiment in the 1st peer review was not added. To be published in Viruses, re-experiments data and discussion are needed to verify. If the authors reconsider the following points in particular, I think that re-submission is possible once the paper is rewritten in accordance with the level expected by Viruses. Other specific comments are as follows.

In the revision of Fig.2, the photographs of CMV symptoms have become easier to see. However, the C and M pictures have been deleted. This deleted data is important. Authors should restore it (like version 1) and add the enlarged picture of V and MV leaf symptoms. Since there is no difference in the viral amount in Fig. 2C, the symptoms photograph in Fig. 2A is extremely important to determine that the DSI in Fig. 2B is correct.

I understood the situation that you could not repeat the experiment because there are no samples left. The authors state their confidence in the data. Authors described that Vitti et al states that SA induction was not observed in tomatoes. I had already checked this reference (Vitti et al 2016) and Fig.6B. This paper showed that total SA content was measured in tomatoes inoculated with the CMV-Fny strain. Therefore, (line 448-449) in the Discussion section "this is not the case in tomato" is not correct. If the authors argue that CMV-infected tomatoes do not induce SA, experiments of proof to overturn the known theory is needed. Authors should try this. If that difficult, at least, authors should perform re-inoculation experiment and SA re-measurement experiment. In addition, authors should be remeasured the level of JA , because Vitti et al also showed the level of JA in tomatoes. The inoculation experiment conducted in this study states that 12 plants were used in each group. However, was there only one times inoculation experiment?

In order to understand the effect of MV more accurately, it is necessary to accurately show the results of control V. Therefore, re-inoculation and re-verification are both required for this paper to meet the criteria for publication in Viruses.

Author Response

In this resubmitted manuscript, the authors changed the results (Abstract: line 29, "without interfering with CMV replication"), and modified the manuscript, and the authors additional experimental analysis of the dry data was performed. However, due to shortage of the stored samples, the wet data of required experiment in the 1st peer review was not added. To be published in Viruses, re-experiments data and discussion are needed to verify. If the authors reconsider the following points in particular, I think that re-submission is possible once the paper is rewritten in accordance with the level expected by Viruses. Other specific comments are as follows. In the revision of Fig.2, the photographs of CMV symptoms have become easier to see. However, the C and M pictures have been deleted. This deleted data is important. Authors should restore it (like version 1) and add the enlarged picture of V and MV leaf symptoms. Since there is no difference in the viral amount in Fig. 2C, the symptoms photograph in Fig. 2A is extremely important to determine that the DSI in Fig. 2B is correct.

Fig.2 has been modified

I understood the situation that you could not repeat the experiment because there are no samples left. The authors state their confidence in the data. Authors described that Vitti et al states that SA induction was not observed in tomatoes. I had already checked this reference (Vitti et al 2016) and Fig.6B. This paper showed that total SA content was measured in tomatoes inoculated with the CMV-Fny strain. Therefore, (line 448-449) in the Discussion section "this is not the case in tomato" is not correct. If the authors argue that CMV-infected tomatoes do not induce SA, experiments of proof to overturn the known theory is needed. Authors should try this. If that difficult, at least, authors should perform re-inoculation experiment and SA re-measurement experiment.

We guess that there was probably a misunderstanding; the focus of our work and manuscript is the effect of mycorrhization on CMV infection in tomato, with particular attention to the mycorrhizal-induced priming phenomenon. We actually wanted to emphasize that in MV plants we detected a higher level of SA, since this is important for the priming phenomenon.

Initially we used the same plant tissue (fifth leaves from the apex) to perform the different experiments in order to analyse the same tissue by different methodologies and get comparable results. However, at the same time we stored apex, first and second leaves at -80°C for further needed experiments and actually we had the opportunity to  repeat the SA and JA evaluation using these samples. Therefore, we were able to obtain a defined quantification of SA in all the considered conditions. The new data confirmed the increase of SA in MV plants and have been added in the text and Fig. 5. Raw data have been inserted as Supplementary material in Table S6.  

Actually, we would like to underling that we didn’t mean to argue about the SA-induction in CMV-infected tomato plants and we won’t overturn the known theory, since this goes beyond the aim of our work.

To avoid any confusion we deleted the related lines (line 448-449) in the Discussion section.

 In addition, authors should be remeasured the level of JA , because Vitti et al also showed the level of JA in tomatoes.

We re-measured the JA levels in new material in parallel with SA but the JA level was always under the detection limit, probably because of limits in sensitivity of the used  analytical method.

This is in agreement with data from Miozzi et al. 2011 where we analysed similar biological material especially concerning the mycorrhizal samples and we found that the physiological level of JA in shoots was under the detection limit while we could detect JA in corresponding root samples.

Moreover, in our opinion the comparison with Vitti et al., 2016 is not fully appropriate since they considered the interaction with plant-CMV and Trichoderma. Trichoderma is a plant endophyte and not a mycorrhizal fungus: the mechanisms of interaction with the host plant and tolerance to pathogens attack could be completely different.  Furthermore, this reference has been deleted from the new revised version of the MS and  is now not present any more.

The inoculation experiment conducted in this study states that 12 plants were used in each group. However, was there only one times inoculation experiment? In order to understand the effect of MV more accurately, it is necessary to accurately show the results of control V. Therefore, re-inoculation and re-verification are both required for this paper to meet the criteria for publication in Viruses.

First of all, we hope that the new experiments that could be performed along with the new data added, essentially confirming previously obtained results and the modifications of the manuscript will meet the favour of the referees and the Editor.

Indeed, we did one inoculation experiment but in order to ensure a good reproducibility level 1) we used an experimental set up already tested in previous works, already published, where we performed analogous experiments on tomato-F. mosseae association under the same controlled experimental and environmental conditions (Maffei et al., 2014); 2) the experiment was done under strictly controlled conditions - that are well established in our laboratories and have led in the past to comparable results in terms of biomass and mycorrhization level; 3) the experiment was based on a number of biological replicates run in parallel that is commonly accepted as sufficient to provide statistical validation to biological analyses.

Concerning the specific point on SA, as requested by the reviewer, we re-measured SA levels in all conditions and changed the results in the new version of the text. The reviewer’s concerns about SA have been considered and discussed above.

To conclude this point, we would like to bring to your attention the fact that several papers already accepted and published in several journals with good IF, report physiological as well as transcriptomic results from biological replicates obtained from one single experiment. Please see the reference list below for examples.

https://doi.org/10.1094/MPMI-03-17-0062-R

https://doi.org/10.1094/MPMI-10-13-0296-R

https://doi.org/10.1094/MPMI-05-14-0126-R

https://doi.org/10.1111/j.1365-313X.2008.03538.x

https://doi.org/10.1111/pce.12902

https://dx.doi.org/10.1038%2Fs41598-019-39463-0

http://www.plantphysiol.org/content/173/4/2180.long

https://onlinelibrary.wiley.com/doi/full/10.1111/pce.13533

https://doi.org/10.1094/MPMI-09-16-0189-R

https://doi.org/10.1094/MPMI-07-16-0146-R

https://doi.org/10.1094/MPMI-05-11-0116